# Differential Regulation of Hyaluronan Synthesis by Three Isoforms of Hyaluronan Synthases in Mammalian Cells

**DOI:** 10.3390/biom14121567

**Published:** 2024-12-09

**Authors:** Jie Wang, Zhikun Wu, Longtao Cao, Feng Long

**Affiliations:** 1Department of Neurosurgery, Zhongnan Hospital of Wuhan University, School of Pharmaceutical Sciences, Wuhan University, Wuhan 430071, China; 2020103060004@whu.edu.cn (J.W.); 2020203060039@whu.edu.cn (Z.W.); 2023203060025@whu.edu.cn (L.C.); 2Ministry of Education Key Laboratory of Combinatorial Biosynthesis and Drug Discovery, School of Pharmaceutical Sciences, Wuhan University, Wuhan 430071, China

**Keywords:** hyaluronan, human hyaluronan synthases, membrane protein expression, enzymatic property, kinetic analysis, hyaluronan-specific ELISA

## Abstract

Hyaluronan (HA) is one of the crucial components of the extracellular matrix in vertebrates and is synthesized by three hyaluronan synthases (HASs), namely HAS1, HAS2, and HAS3. The low expression level of HASs in normal keratinocytes and other various types of cells presents a recognized challenge, impeding biological and pathological research on their localization. In this study, the human proteins HAS1, HAS2, and HAS3 with fused maltose-binding protein (MBP) tags were successfully expressed at high levels and purified for the first time in HEK293F cells. The enzymatic properties of the three HAS proteins were further characterized and compared. A pulse-field gel electrophoresis analysis of the size distribution of the hyaluronan generated in vitro by the membrane proteins demonstrated that the three HAS isoforms generate HA polymer chains at different molecular masses. Kinetic studies demonstrated that the three HAS proteins differed in their catalytic efficiency and apparent K_m_ values for the two substrates, UDP-GlcA and UDP-GlcNAc. Furthermore, the cellular hyaluronan secretion by the three isoenzymes was evaluated and quantified in the HEK 293T cells transfected with GFP-tagged HAS1-GFP, HAS2-GFP, and HAS3-GFP using an ELISA assay. These findings enhance our understanding of the membrane protein HASs in mammalian cells.

## 1. Introduction

Hyaluronan (hyaluronic acid, HA) is a linear glycosaminoglycan (GAG) of the extracellular matrix (ECM), which provides mechanical and physiological support to the surrounding cells [1]. HA not only acts as an important structural element in cartilage, synovial fluid, the vitreous humor of eye, and the skin of vertebrates [2,3], but also plays a crucial role in many physiological processes such as cell adhesion, migration and differentiation [4,5], tumor invasion and metastasis [6,7], wound healing, inflammation, and angiogenesis [8,9]. Due to its distinctive moisture retention ability and viscoelastic property, as well as its lack of immunogenicity and toxicity, HA has a wide range of applications in the cosmetics, biomedical, and food industries [10,11,12].

The chemical structure of HA is a ubiquitous linear glycosaminoglycan comprising of alternating uridine diphosphate glucuronate (UDP-GlcA) and uridine diphosphate N-acetylglucosamine (UDP-GlcNAc) residues, which are linked via *β*-1,4 or *β*-1,3 glycosidic bonds [13,14] (Figure 1A). Unlike the other glycosaminoglycans mainly synthesized at the Golgi apparatus, HA is synthesized on the plasma membrane by a group of enzymes called hyaluronan synthases (HASs) [15]. HASs have been divided into two classes. In eukaryotic and streptococcal species, HA is synthesized by the Class I HASs, which are lipid-dependent proteins with integral multi-membrane-spanning regions. In contrast, the Class II HASs are bifunctional cytosolic glycosyltransferases and are only found in bacteria [16]. To date, three Class I HAS isozymes, namely HAS1, HAS2, and HAS3, have been identified in mammals [17]. These HASs exhibit unique glycosyltransferase activity, which includes initiating HA biosynthesis using only UDP-GlcNAc, UDP-GlcA, and Mg^2+^ as the major sources, extending the nascent HAs by adding new intracellular sugar-UDPs at the reducing end of the growing hyaluronyl-UDP chains, and extruding the HA products out of the cells [18] (Figure 1B).

In mammalian cells, the HAs generated by HAS1, HAS2, and HAS3 differ in size distribution and tissue specificity [19]. In general, the length of HA molecules can range from oligomers to extremely long forms, reaching up to millions of Daltons in molecular weight, and the HA biological functions are closely associated with its molecular mass. High-molecular weight HA contributes to tissue homeostasis [20], displaying anti-inflammatory [21] and antioxidant properties [22]. Under physiological conditions, most extracellular HAs exhibit a relatively high molecular weight. In contrast, low-molecular weight HA is linked to inflammation, tissue injury, and cancer metastasis [23,24,25]. Therefore, synthesizing HAs of different sizes can potentially impact cell behavior. However, the mechanisms related to how the mammalian isoforms HAS1, HAS2, and HAS3 work and coordinate for proper biophysiological functions in cells remain unclear.

In this study, we biochemically characterized three human transmembrane enzymes HAS1, HAS2, and HAS3. The heterogenous protein expression of the human HASs was optimized in eukaryotic HEK293F cells. The enzymatic properties of the human HASs and their products were further examined in vitro. Our investigation revealed that the three HAS enzymes exhibited significant differences in catalytic efficiency, HA elongation rate, hyaluronan secretion level, and K_m_ parameters for the substrates UDP-GlcA and UDP-GlcNAc.

## 2. Materials and Methods

### 2.1. Transient Expression of Human HAS1, HAS2, and HAS3 in HEK293F Cells

The cDNAs for human HAS1 (Uniprot: Q92839), HAS2 (Uniprot: Q92819), and HAS3 (Uniprot: O00219) were codon-optimized and synthesized by a commercial source (GenScript, Nanjing, China). They were separately cloned into the pEGBacMam vector, with an N-terminal Flag tag, 8 × His tag, maltose-binding protein, and a tobacco etch virus (TEV) cleavage site, and expressed in HEK293F cells (Gibco) using the BacMam system (Thermo Fisher Scientific, Boston, MA, USA) [26]. The baculoviruses were generated using the Bac-to-Bac system and amplified in Sf9 cells (Invitrogen, Carlsbad, CA, USA). The HEK293F cells were cultured in an SMM 293T-II medium (Sino Biological Inc., Beijing, China) supplemented with 8% CO_2_ in a Multitron-Pro shaker (Infors, Basel, Switzerland, 125 r.p.m.) at 37 °C. When the cell density reached 5 × 10^6^ cells per mL, the culture was diluted to 3 × 10^6^ cells per mL in a fresh medium and transfected with the P3 viruses at a ratio of 1 to 10 (virus HEK293F, *v*/*v*). The infected cells were cultured in suspension at 37 °C for 12 h and supplemented with 10 mM of sodium butyrate to facilitate protein expression. The cells were then cultured in suspension at 37 °C for 60 h, collected by centrifugation at 5000 rpm for 15 min at 4 °C, and stored at −80 °C until purification.

### 2.2. Protein Purification of Human HAS1, HAS2, and HAS3

To purify HAS1, the cell pellets were resuspended and lysed in a lysis buffer (25 mM HEPES (pH 7), 150 mM NaCl, 10% (*v*/*v*) glycerol) supplemented with 5 mM of MgCl_2_ and 1 mM of phenylmethanesulfonyl fluoride (PMSF). The cell membrane was collected by ultracentrifugation (140,000× *g*, 1.5 h, 4 °C) and solubilized with a lysis buffer supplemented with 1% Lauryl Maltose Neopentyl Glycol (LMNG, Anatrace), 0.1% Cholesteryl Hemisuccinate Tris Salt (CHS, Anatrace), 2 μg/mL of aprotinin, 1 μg/mL of pepstatin, and 1 μg/mL of leupeptin at 4 °C for 4 h. After centrifugation at 140,000× *g* for 1.5 h at 4 °C, the supernatant was subjected to affinity chromatography using the Anti-Flag affinity resin and incubated for 1 h at 4 °C with shaking. The resin was washed twice with 30 mL of a high-osmotic buffer containing 25 mM of HEPES (pH 7), 500 mM of NaCl, 0.02% (*w*/*v*), and glyco-diosgenin (GDN, Anatrace, Maumee, OH, USA), followed by 20 mL of a low-osmotic buffer containing 25 mM of HEPES (pH 7), 150 mM of NaCl, and 0.02% (*w*/*v*) GDN. The target proteins were eluted with a low-osmotic buffer plus 300 μg/mL of a Flag peptide. The elution fractions were then concentrated to approximately 4.3 mg/mL (300 μL) using a 50 kDa cut-off Amicon (Millipore, Burlington, VT, USA).

The protocols for purifying the HAS2 and HAS3 proteins were nearly identical to that for the membrane protein HAS1. For the purification of HAS2 and HAS3, the membrane fractions were resuspended and solubilized in 25 mM of HEPES (pH 8), 150 mM of NaCl, and 10% glycerol, supplemented with 1% n-Dodecyl-β-d-Maltopyranoside (DDM, Anatrace) and 0.1% CHS, then incubated for 4 h with agitation. The proteins were eluted from the Anti-Flag affinity resin with a buffer containing 25 mM of HEPES (pH 8.0), 150 mM of NaCl, 300 μg/mL of a Flag peptide, and 0.02% GDN. Subsequently, the elution fractions of HAS2 and HAS3 were concentrated to approximately 4.0 mg/mL (300 μL) and 5.7 mg/mL (300 μL), respectively, using a 50 kDa cut-off Amicon.

### 2.3. Synthesis of HA Polysaccharides In Vitro

The reaction was carried out in 20 mM of HEPES (pH 7.5), 150 mM of NaCl, and 0.02% GDN at 37 °C for 3 h in the absence of light and in a total volume of 20 μL. The reaction mixture contained 2.5 mM of both UDP-GlcA and UDP-GlcNAc, and 20 mM of MgCl_2_. The reaction was initiated by the addition of the recombinant proteins HAS1, HAS2, and HAS3 to the final estimated concentrations of 15 μM, 10 μM, and 15 μM, respectively.

### 2.4. HA Analysis by Pulse-Field Gel Electrophoresis

The synthesized HA was mixed with a sucrose solution (final concentration of 500 mM) and loaded onto a 0.7% agarose gel. Electrophoresis was performed at 60 V for 5 h with pre-cooling in a TBE buffer. After the run, the gel was equilibrated in 50% ethanol for 1.5 h and incubated in 0.01% Stains-All and in 50% ethanol overnight in the dark. The post-staining background was reduced by soaking the gel in 20% ethanol for two days in the dark. Finally, gel images were collected under white light transillumination using a ChemiDoc XRS + system (Bio-Rad, Hercules, CA, USA).

### 2.5. Enzyme-Coupled Activity Assay

The kinetic parameters of HAS1, HAS2, and HAS3 were obtained using the pyruvate kinase (PK)-lactate dehydrogenase (LDH)-coupled photometric assay in the Appendix A. The reaction mixture contained 25 mM of HEPES (pH 8.0), 150 mM of NaCl, 20 mM of MgCl_2_, 1 mM of phosphoenolpyruvate (PEP), 0.75 mM of reduced β-nicotinamide adenine dinucleotide (NADH), 4U LDH, 3U PK, 2.5 mM of either UDP-GlcA or UDP-GlcNAc, and the indicated amounts of UDP-GlcA or UDP-GlcNAc in a 100 µL total volume, assayed in 96-well microtiter plates (Corning, Corning, CA, USA). HA synthesis was initiated by adding 10 µL of HASs to the reaction mix, and readings were taken at 10 min intervals at 340 nm for 2 h at 37 °C using a SpectraMax iD3 instrument (Molecular Devices, San Jose, CA, USA). The rate of NADH depletion was converted to UDP and released for a Michaelis–Menten constant determination using GraphPad Prism (GraphPad Software, San Diego, CA, USA). The K_m_ values for the two substrates, UDP-GlcA and UDP-GlcNAc, were determined by measuring the synthase activity as a function of the UDP-sugar concentration. Double-reciprocal plots of 1/V versus 1/S gave the V_max_ values of the recombinant proteins HAS1, HAS2, and HAS3 for UDP-GlcA and UDP-GlcNAc. All of the experiments were performed in triplicate.

### 2.6. ELISA Assay of Hyaluronan

For the measurement of hyaluronan secretion into the culture media, 4 × 10^4^ HEK293T cells were plated in each well of the 24-well plates (Corning) and transfected the next day with HAS1-GFP, HAS2-GFP, and HAS3-GFP constructs. A day after transfection, 2 mM of UDP-GlcA (Sigma, St. Louis, MO, USA), 2 mM of UDP-GlcNAc (Sigma), and 20 mM of MgCl_2_ (sigma) were added to the fresh culture medium as substrates. After 48 h of incubation, the supernatant was collected, and the cells were counted. The quantification of the HA secreted into the growth medium was performed using the commercial enzyme-linked immunosorbent assay (ELISA) Human Hyaluronic acid HM10597 Kit (Bioswamp, Wuhan, China), according to the manufacturer’s instructions, utilizing a biotinylated hyaluronan-binding complex (bHABC) prepared in the laboratory.

## 3. Results

### 3.1. Bioinformatics Analysis of HAS1, HAS2, and HAS3

In order to understand the evolutionary status of HASs, a phylogenetic analysis was conducted based on the amino acid sequences encoded by HAS genes from 30 different sources, which were separated into four groups, including HAS1, HAS2, and HAS3 from vertebrates, as well as those from bacteria and viruses. A rooted phylogenetic tree was generated through the neighbor-joining method, which visually presents the phylogenetic relationships between these proteins (Figure 2A). The amino acid sequences of three human HAS1, HAS2, and HAS3 were aligned using the Clustal Omega program, revealing a high sequence similarity of 66% (Figure 2B). HAS1, HAS2, and HAS3 are all complex membrane proteins with 6–7 predicted transmembrane (TM) domains and an intracellular catalytic domain (Figure 1C).

### 3.2. Expression and Purification of Recombinant Human HAS1, HAS2, and HAS3

Because of the low expression levels of human HASs in various cell types, the expression and purification of these proteins for subsequent activity studies has been limited. The maltose-binding protein (MBP) moiety is commonly utilized to enhance the solubility and promote the proper folding of its fusion partners [27,28]. To enhance the expression and solubility of the human HASs in HEK293F cells, fusion constructs were created by appending a maltose-binding protein (MBP) tag to the N-terminus of HAS1, HAS2, and HAS3, respectively. The recombinant protein of HAS1-MBP was extracted from the membrane fraction using the detergent Lauryl Maltose Neopentyl Glycol (LMNG) and purified using Anti-Flag affinity chromatography in the buffer containing glyco-diosgenin (GDN) detergent. Following Anti-Flag resin affinity chromatography, HAS1-MBP was obtained in the elution buffer containing 300 μg/mL of Flag peptide. The analysis of HAS1-MBP by sodium dodecyl sulfate-polyacrylamide gel electrophoresis (SDS-PAGE) depicted a single band with a molecular weight of about 108 kDa (Figure 3A), which was consistent with the expected size of the recombinant protein; the original figures can be found in the Appendix A.

Similarly, the recombinant proteins HAS2-MBP and HAS3-MBP were initially solubilized from the membrane fraction in the detergent n-Dodecyl-β-d-Maltopyranoside (DDM) and exchanged into GDN afterwards during purification. After SDS-PAGE gel electrophoresis, two protein bands at around 100 kDa were observed in the samples, corresponding to the protein sizes of HAS2 (107 kDa, Figure 3B) and HAS3 (106 kDa, Figure 3C), respectively; the original figures can be found in the Appendix A. During the experimental process, it was observed that HAS1 exhibited greater stability during purification when extracted with the LMNG detergent. Conversely, HAS1 showed more pronounced aggregation and precipitation during purification after its extraction by DDM. Moreover, the expression of the recombinant HAS1-MBP, HAS2-MBP, and HAS3-MBP proteins was confirmed by a Western blot analysis using the Anti-His-tag and Anti-Flag-tag antibodies (Figure 3D); the original figures can be found in the Appendix A. These results indicate that, in heterologous expression systems using an equal scale for cell cultures, the expression level of the recombinant membrane protein HAS3 is generally slightly higher than that of HAS1 and HAS2. This observation is consistent with the distinct expression patterns of the three HAS isoforms, where HAS3 is typically expressed at higher levels in certain physiological and pathological conditions. For example, HAS3 expression is significantly increased in certain cancers, which contributes to the metastatic potential of tumor cells. Thus, the higher expression of HAS3 in these systems may reflect its role in more dynamic cellular processes, such as tumor progression and metastasis [29,30,31].

### 3.3. The Concentration of HA Polymer Synthesis Analyzed by Pulse-Field Gel Electrophoresis

The three human HAS isoforms exhibit different biochemical properties in terms of their catalytic efficiency in synthesizing hyaluronan in vitro [32]. To assess the catalytic activity of HAS1, HAS2, and HAS3, the synthetic HA generated by our purified HAS isoforms was subjected to pulse-field gel electrophoresis followed by Stains-All staining. We investigated the concentrations of each HAS protein required for HA synthesis under the saturated conditions of the supplied substrates UDP-GlcA and UDP-GlcNAc, using various concentrations of HASs (0, 1, 3, 5, 10, 15, 20, 30 μM). The agarose gel analysis of the synthesized HA revealed that the length of the synthesized HA polymer gradually increased with the increasing concentrations of HAS1 (Figure 4A), HAS2 (Figure 4B), and HAS3 (Figure 4C); the original figures can be found in the Appendix A. These polysaccharides synthesized by hyaluronan synthases were easily degraded by hyaluronidases, confirming their authenticity as genuine HA. The length of the synthesized HA polymers remained constant in the agarose gel until the concentrations of HAS1, HAS2, and HAS3 reached 15 μM (Figure 4D), 10 μM (Figure 4E), and 15 μM (Figure 4F), respectively.

### 3.4. Size Distribution of HA Synthesized by HAS1, HAS2, and HAS3

Although all HAS isoforms (HAS1-3) produce HA, the three different human HAS isozymes synthesize HA of different sizes, each with distinct biological significance. The analysis using agarose gel electrophoresis indicated that HAS1 and HAS2 polymerize the migrating HA chains with average molecular mass of approximately 2 × 10^6^ Da above a 1.6 × 10^6^ Da HA standard marker, while HAS3 produced shorter chains ranging from 1 × 10^6^ Da to 2 × 10^6^ Da (Figure 4G); the original figures can be found in the Appendix A. Notably, among them, HAS2 produced high-molecular weight HA at a relatively higher rate compared to the other HAS isoforms. Therefore, HAS2 is probably the primary enzyme responsible for regulating the concentration of high-molecular weight HA in tissues.

### 3.5. Monitoring the Progression of In Vitro HA Biosynthesis over Time

We further examined whether the molecular size of HA was affected by the incubation times. The cultivation time for synthesizing HA by HAS2 was investigated due to its high production rate of high-molecular weight HA. The glycosyltransferase HAS incorporates glucuronic acid and N-acetylglucosamine into alternating positions in the chain, utilizing UDP-glucuronic acid (UDP-GlcA) and UDP-N-acetylglucosamine (UDP-GlcNAc) as substrates. Consequently, as the incubation time increases, the size of HA synthesized by HAS2 gradually increases as well. Agarose gel electrophoresis revealed that the molecular size of the HA generated by HAS2 showed a gradual increase in polymer lengths until 2 h, synthesizing the HA polymers of a consistent length that were migrating above a 2 × 10^6^ Da standard marker (Figure 4H); the original figures can be found in the Appendix A. The result demonstrated that the HAS enzymes possessed an intrinsic ability to regulate the size of the HA produced.

### 3.6. Sensitivity of HAS2 to UDP-GlcA and UDP-GlcNAc Substrates

The HAS enzymes alternatively transfer glycosyl groups from UDP-GlcA or UDP-GlcNAc into the growing HA chain, with the rate of hyaluronan synthesis depending on the supply of these substrates. The recombinant human HAS2 was co-incubated with various concentrations (0, 0.5, 1, 2.5, 5.0 mM) of UDP-GlcA or UDP-GlcNAc, while maintaining the concentration of the other substrate saturated at 2.5 mM. The result indicated that even at low concentration (0.5 mM) of either UDP-GlcA or UDP-GlcNAc, the produced HA molecules had molecular weights exceeding 2 × 10^6^ Da (Figure 4I); the original figures can be found in the Appendix A. Therefore, in vivo, HA biosynthesis may stall due to substrate depletion, competitive UDP inhibition, or HA accumulation.

### 3.7. Kinetic Analyses of Membrane Proteins HAS1, HAS2, and HAS3

The enzyme-coupled activity assay is used for the investigation of the kinetics and mechanisms of HASs [33]. Under constant near-saturation substrate concentrations, a linear correlation was observed between the rate of HA synthesis and the enzyme concentrations of HAS1, HAS2, and HAS3 (Figure 5).

The Michaelis–Menten constants (K_m_ values) for the two substrates, UDP-GlcA and UDP-GlcNAc, were estimated by monitoring the release of UDP during the polymerization reaction using an enzyme-coupled reaction that oxidizes NADH (Appendix A). The results revealed that the apparent K_m_ values of UDP-GlcA and UDP-GlcNAc for the membrane protein HAS1 were approximately 0.44 mM and 0.84 mM, respectively (Figure 6A). The membrane protein HAS2 exhibited K_m_ values of around 0.27 mM and 0.43 mM for UDP-GlcA and UDP-GlcNAc, respectively (Figure 6B). Similarly, the apparent Michaelis constants, K_m_, for the UDP-GlcA and UDP-GlcNAc of the membrane protein HAS3 were about 0.32 mM and 0.74 mM, respectively (Figure 6C). The three proteins HAS1, HAS2, and HAS3 exhibited a gradual increase in the synthesis rate of HA with increasing substrate concentrations up to 1.5–2 mM. With a sufficient reaction time, when the concentration of one substrate (either UDP-GlcA or UDP-GlcNAc) becomes oversaturated, the kinetic analysis of the three membrane proteins is performed with respect to the other substrate. The membrane protein HAS2 exhibited lower K_m_ values for both UDP-GlcA and UDP-GlcNAc than those of HAS1 and HAS3, indicating that HAS2 has a higher affinity for UDP-GlcA and UDP-GlcNAc than HAS1 and HAS3. Additionally, all the HAS enzymes exhibited a lower affinity for UDP-GlcNAc than for UDP-GlcA. Furthermore, the V_max_ values of the membrane proteins HAS1, HAS2, and HAS3 did not show significant differences (Figure 6D).

### 3.8. HAS1-GFP, HAS2-GFP, and HAS3-GFP Overexpressing 293T Cells Secreted Hyaluronan

The human HAS1, HAS2, and HAS3 with C-terminally fused green fluorescence proteins (GFPs) were transiently expressed in the mammalian HEK293T cells. In the cells transfected with equal amounts of plasmids, HAS3-GFP exhibited a higher protein expression level compared to HAS1-GFP and HAS2-GFP, as indicated by the overall enhanced fluorescence intensity of the positive cells (Figure 7A). The expression of the HAS1-GFP, HAS2-GFP, and HAS3-GFP proteins was confirmed by a Western blot analysis using the Anti-His-tag and Anti-Flag-tag antibodies (Figure 7B); the original figures can be found in the Appendix A. The results showed that the expression level of HAS3-GFP is slightly higher than that of HAS1-GFP and HAS2-GFP, which is consistent with what we observed in the heterogenous expression of HASs-MBP. The secreted HA in the supernatant of the transfected 293T cell culture was measured and compared using an HA-specific ELISA after culturing it for 48 h without substrates and with the addition of 2 mM of UDP-GlcA and 2 mM of UDP-GlcNAc. With the additional substrates in the culture medium, HAS1-GFP, HAS2-GFP, and HAS3-GFP exhibited moderate increases in hyaluronan secretion compared to the condition without additional substrates (Figure 7C). The HA-specific ELISA results indicated that HAS2-GFP secreted more hyaluronan than HAS1-GFP and HAS3-GFP.

## 4. Discussion

Mammalian genomes have three different HAS genes (HAS1, HAS2, and HAS3) which are expressed at specific stages and in specific tissues during organ development, aging, wound healing, as well as under normal or pathologic conditions, including diseases such as cancer [34]. However, the relatively low expression levels of HAS in many cell types hinder research into its role in biological and pathological processes. In our present study, the human HAS1, HAS2, and HAS3 were fused with a maltose-binding protein (MBP) tag and effectively expressed in significant quantities in HEK293F cells for the first time. The MBP tag enhances protein expression and solubility and facilitates proper folding. To improve the expression of the recombinant proteins HAS1, HAS2, and HAS3, the BacMam system was employed using a combination of mammalian HEK293F cells and insect Sf9 cells. Baculovirus production and amplification were performed using Sf9 cells, followed by recombinant protein expression in the HEK293F cells. The solubilization of hyaluronan synthase from the crude membrane fractions of eukaryotic cells with detergents was attempted to purify the proteins. It has been shown that Triton X-100 was a useful detergent for solubilizing bacterial HAS from *Streptococcus equisimilis* in its active form [16]. Recently, the mild detergents DDM and LMNG were identified as effective detergents for solubilizing and purifying *Chlorella viruses* HAS and *Xenopus laevis* HAS proteins in their active form [35,36]. We found that the recombinant protein HAS1-MBP preferred different detergents to maintain its stability in a solution when compared to HAS2-MBP and HAS3-MBP.

HA, cellulose, and chitin synthases are multifunctional enzymes that synthesize high-molecular-weight extracellular polysaccharides [37]. However, the mechanisms through which these enzymes regulate product lengths and, thus, the physical properties of the polymers are largely unresolved. Our present study showed that the HAS isoforms synthesized HA at various molecular weights. HAS1 and HAS2 polymerized the HA chains of a weight greater than approximately 2 × 10^6^ Da, while HAS3 formed shorter chains ranging from 1 × 10^6^ Da to 2 × 10^6^ Da. It was also interesting to observe that HAS2 was required at a lower concentration for HA synthesis compared to HAS1 and HAS3, while exhibiting a relatively higher rate of production of high-molecular weight HA. Overall, the observed product size distributions synthesized by HAS1, HAS2, and HAS3 reflect the HA size range produced by mammals’ HAS isoenzymes in vitro.

Comparing the enzymatic properties of the three mammalian hyaluronan synthases, HAS1, HAS2, and HAS3 may provide insights into their respective roles in HA biosynthesis and elucidate their functional relationships. Weigel and colleagues showed that the two bacterial hyaluronan synthases from *Streptococcus pyogenes* and *Streptococcus equisimilis* have different enzymatic activities and kinetic behaviors [38]. Although all three HAS isoforms share similarities in protein sequence and function, some differences have been noted in enzyme catalytic efficiency, chain elongation rate, chain length, and substrate-dependent enzyme properties [39]. We compared the differences in kinetic behavior of the three recombinant HAS proteins using an enzyme-coupled activity assay. The apparent K_m_ values of the HAS proteins for the substrates UDP-GlcA and UDP-GlcNAc were determined by assessing the synthase activity at varying concentrations of UDP-sugars. The membrane protein HAS2 exhibited a lower K_m_ value for both UDP-GlcA and UDP-GlcNAc than HAS1 and HAS3. Therefore, HAS2 demonstrated a higher affinity for UDP-GlcA and UDP-GlcNAc compared to HAS1 and HAS3. The finding indicated that efficient hyaluronan synthesis requires higher cellular concentrations of UDP-GlcA and UDP-GlcNAc in the order of HAS1 > HAS3 > HAS2. Among the three isoforms, HAS2 might play a major role in HA production as it relates to cellular function. By generating inducible HAS1-GFP, HAS2-GFP, and HAS3-GFP in the overexpressing 293T cell line, we were able to measure the secretion of hyaluronic acid in the culture medium using a specific hyaluronan ELISA. The concentration of hyaluronan in the medium was significantly higher in the cultures transfected with the HAS1-GFP, HAS2-GFP, and HAS3-GFP constructs compared to the culture transfected with the empty GFP vector. The secretion of hyaluronan increased after the addition of 2 mM of UDP-GlcA and 2 mM of UDP-GlcNAc, compared to the condition without a substrate addition.

Naoki and colleagues showed that the three isoforms of hyaluronan synthase in mice exhibit distinct enzymatic activities [40]. In their study, the COS-1 cells were used for the low-level expression of HAS proteins, and enzyme stability, the HA extension rate, and the kinetics of the two substrates UDP-GlcNAc and UDP-GlcUA were analyzed. Similarly, in our study, we performed an enzyme analysis on the three human hyaluronan synthase isoforms, evaluating enzyme stability, the elongation rate of HA, chain length, and the substrate-dependent kinetics. The key difference in our research was that we investigated the large-scale expression of three recombinant human hyaluronan synthases in HEK293F cells, followed by the in vitro purification of the membrane proteins. Additionally, we quantified the hyaluronan levels in the culture medium using a specific hyaluronan ELISA.

Furthermore, while the in vitro conditions enabled us to assess the activity of the membrane proteins, they may not have fully replicated their behavior in native membrane environments. The kinetic data obtained from the in vitro experiments, as well as the size of the synthesized HA, may differ from what occurs under in vivo physiological conditions. Thus, in the future, further investigation of HASs under physiological conditions may be useful.

In conclusion, we successfully expressed and purified the human HAS1, HAS2, and HAS3 proteins using a eukaryotic expression system and validated their enzymatic activity in vitro. The three mammalian HA synthases are related to each other but exhibit distinct enzymatic properties, suggesting diverse physiological roles for each synthase. This provides a necessary foundation for future research focused on analyzing the structural and functional characteristics of HAS proteins.

## Figures and Tables

**Figure 1 biomolecules-14-01567-f001:**
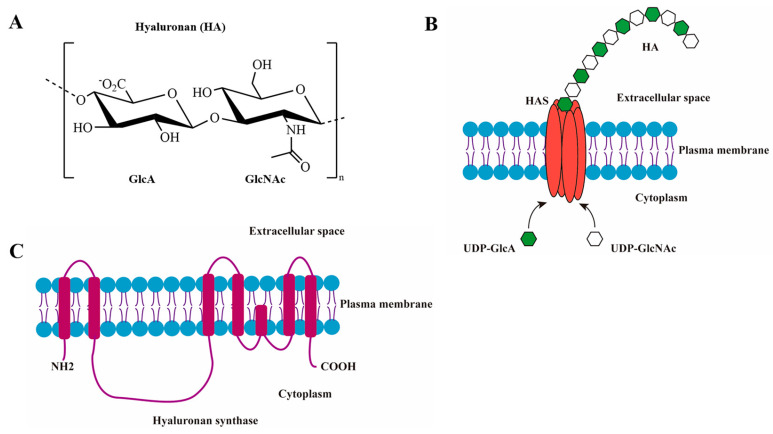
Biosynthesis of hyaluronan. (**A**) Chemical structure of an HA disaccharide unit. (**B**) Schematic illustration of HA synthesis and secretion. (**C**) Predicted structure of mammalian HAS.

**Figure 2 biomolecules-14-01567-f002:**
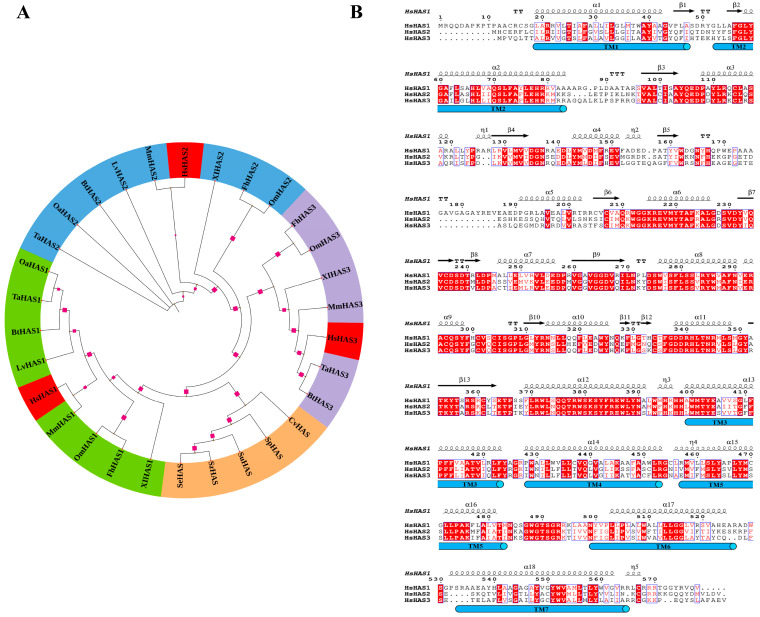
Phylogenetic analysis and Sequence homology of HASs. (**A**) Phylogenetic analysis of HASs from vertebrates, bacteria, and viruses. A phylogenetic tree was constructed using the neighbor-joining method based on the protein sequences of 30 HAS genes. (**B**) Sequence alignment and secondary structure of three human HAS isoforms. The multiple sequence alignments for human HAS1 (HsHAS1), human HAS2 (HsHAS2), and human HAS3 (HsHAS3) were generated using Clustal Omega and displayed using ESPript 3.0.

**Figure 3 biomolecules-14-01567-f003:**
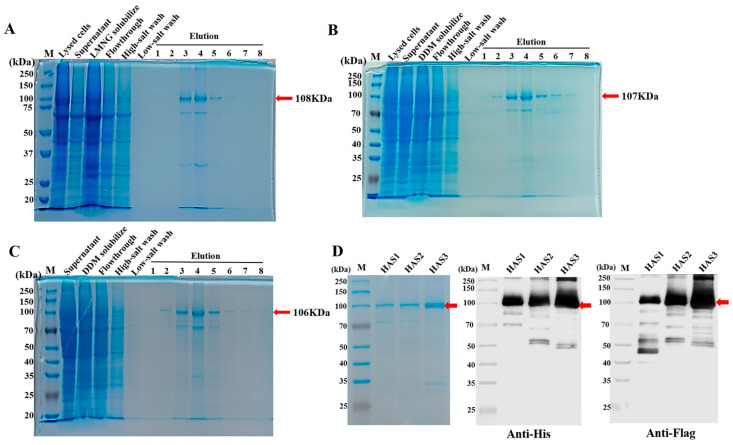
The expression and purification of the recombinant HAS proteins. (**A**–**C**) The SDS-PAGE analysis evaluated the purified HAS1-MBP (**A**), HAS2-MBP (**B**), and HAS3-MBP (**C**) using the eukaryotic expression system. (**D**) The Western blot analysis examined the expression of the recombinant HAS1-MBP, HAS2-MBP, and HAS3-MBP proteins using the Anti-His-tag and Anti-Flag-tag antibodies.

**Figure 4 biomolecules-14-01567-f004:**
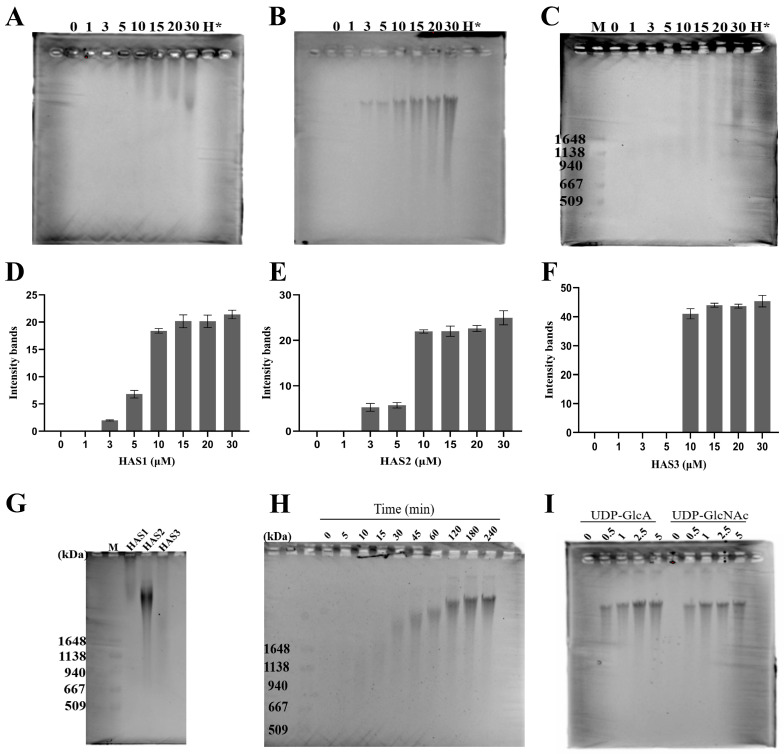
HA biosynthesis in vitro monitored by agarose gel electrophoresis and Stains-All staining. (**A**–**C**) Utilizing different concentrations (0, 1, 3, 5, 10, 15, 20, 30 μM) of HAS1 (**A**), HAS2 (**B**), and HAS3 (**C**) for regulated HA synthesis. H*: HA synthesized from 20 μM of HAS digested by hyaluronidase. (**D**–**F**) Quantitative analysis of the gel images, corresponding to Figure 4A–C, respectively, was performed using Image J software (version 1.53r, National Institutes of Health, Bethesda, MD, USA) and the plots were generated with GraphPad software (version 9.0, San Diego, CA, USA). Error bars represent SD. (**G**) Size distribution of HA synthesized by mammalian HAS proteins. M, the molecular weight marker represents HA size standards. (**H**) Monitoring the time course of HA synthesis by HAS2 in vitro using agarose gel electrophoresis. (**I**) Catalytic activity depends on the concentration of the substrates UDP-GlcA and UDP-GlcNAc.

**Figure 5 biomolecules-14-01567-f005:**
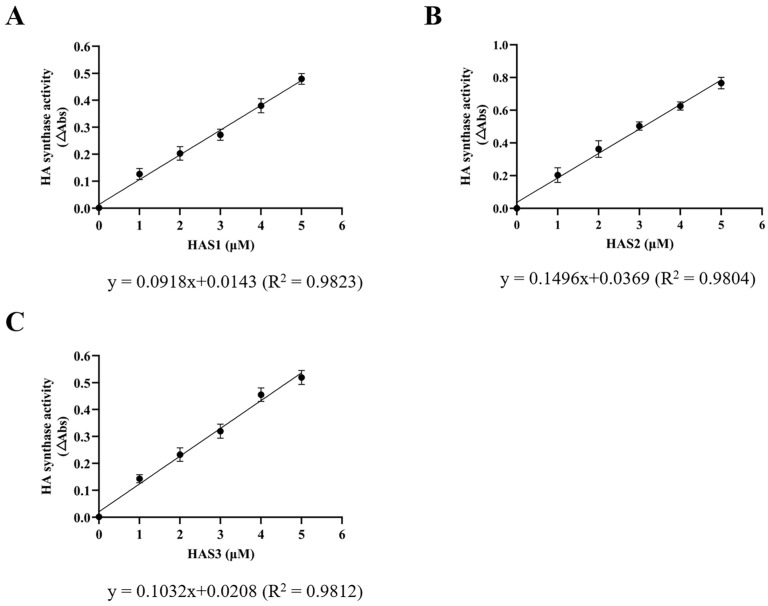
Determining the relationship between the HA synthesis rate and the concentrations of HAS1, HAS2, and HAS3 by kinetic analysis. The data were obtained under the saturating conditions of two substrate concentrations, UDP-GlcA and UDP-GlcNAc. The HA synthesis rate exhibited a linear relationship with the concentrations of HAS1 (**A**), HAS2 (**B**), and HAS3 (**C**), respectively.

**Figure 6 biomolecules-14-01567-f006:**
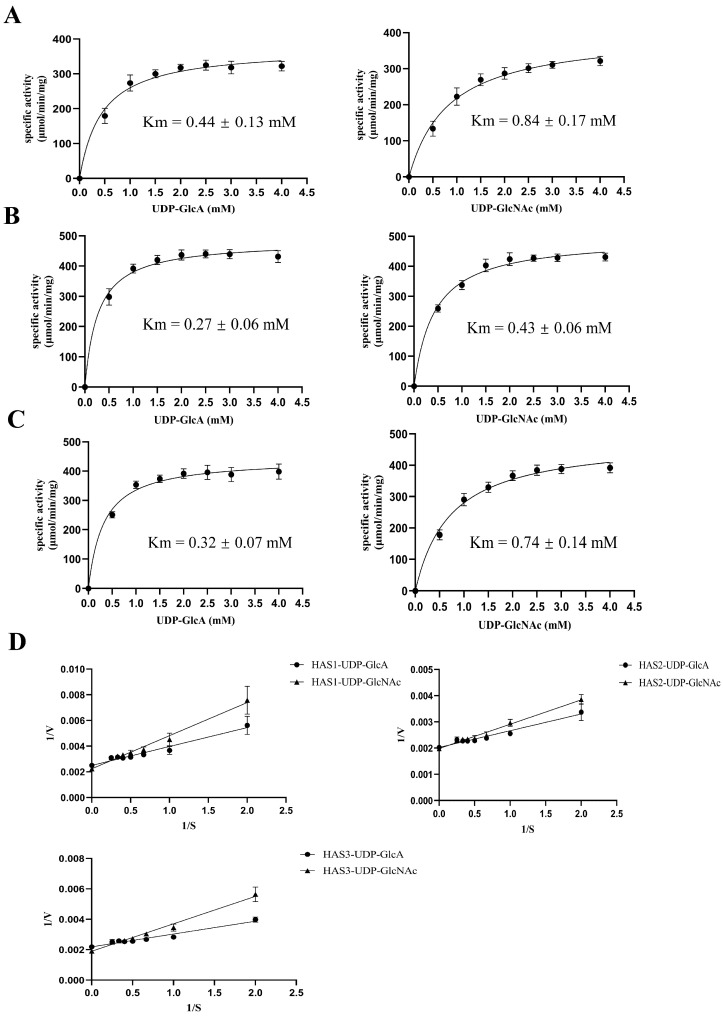
Kinetic analysis of three HAS isozymes. Release of UDP was quantified in real time using an enzyme-coupled assay that monitors the oxidation of NADH. (**A**–**C**) Either UDP-GlcA or UDP-GlcNAc was titrated while the concentration of the other substrate was 2.5 mM. Kinetic parameters for HAS1 are presented in (**A**), for HAS2 in (**B**), and for HAS3 in (**C**). (**D**) Double-reciprocal plot estimation of V_max_ for UDP-GlcA and UDP-GlcNAc. The specific incorporation data used to generate (**A**–**C**) were plotted as 1/V versus 1/S.

**Figure 7 biomolecules-14-01567-f007:**
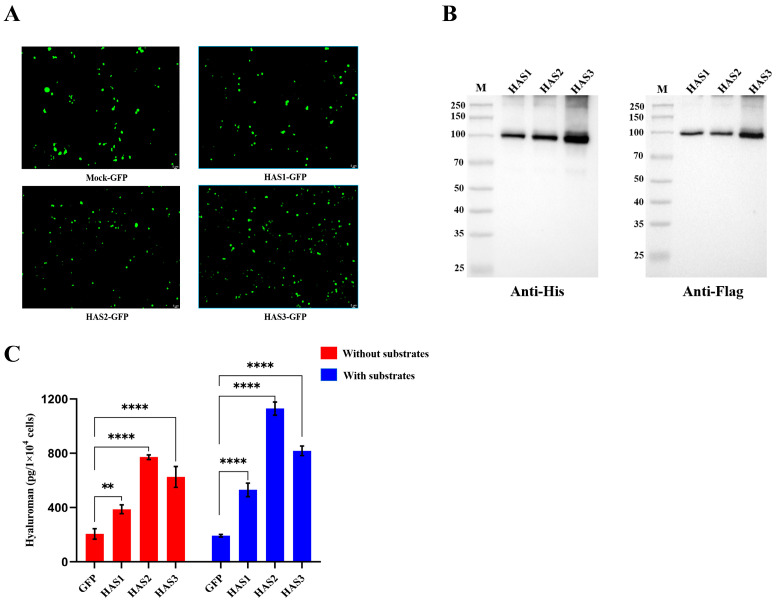
Hyaluronan secretion of HEK293T cells transfected with constructs expressing HAS-GFP fusion proteins. (**A**) The fluorescence intensity of GFP control, HAS1-GFP, HAS2-GFP, and HAS3-GFP fusion proteins transfected into HEK293T cells. (**B**) The Western blot analysis examined the expression of the HAS1-GFP, HAS2-GFP, and HAS3-GFP proteins using the Anti-His-tag and Anti-Flag-tag antibodies. (**C**) Hyaluronan-specific ELISA assay-based quantification of secreted HA. Red: without substrates addition. Blue: the addition of 2 mM of UDP-GlcA and 2 mM of UDP-GlcNAc. Error bars represent deviations from the means with *n*  =  3 in the independent experiments, the asterisks indicate a *p*-value of <0.01 (**), or <0.0001 (****).

## Data Availability

All data are contained in this article.

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
