# Peer review of "Differential Regulation of Hyaluronan Synthesis by Three Isoforms of Hyaluronan Synthases in Mammalian Cells"

_biomolecules, 2024, doi:10.3390/biom14121567_

Round 1

Reviewer 1 Report

Comments and Suggestions for Authors

Wang et al have studied and optimized the expression of isoforms of Hyaluronan synthase (HAS1, HAS2, and HAS3) and also conducted kinetic studies. Finally, authors quantified the expression of three homologous using ELISA. Overall the study is comprehensive, structured and provide a major step in our understanding the hyaluronan biosynthesis which is indispensable and play crucial role in dynamic cellular processes. But shortcomings in the results should be addressed before the accepting it for publication.

1.       In figure 3 (A-C): Why does author not purify HAS based on their sizes (eg. size exclusion chromatography) after affinity chromatography to remove the nonspecific protein bands seen in the SDS PAGE gels and conducted all the further studies with that fraction of purified protein?

2.       In figure 3D: The western blot quality is very low for both anti-His and anti-Flag. Authors should provide western blot of higher quality or may use anti-MBL antibody. (Ref: Goentzel et al, 2006)

3.       For kinetic studies, authors should check and report the kinetics of at least one of the isoforms as a control after removing the MBP protein, if solubility does not compromise.

4.       In figure 4A and 4C: the quality of the gels are very poor and the results deduced from the gels are not validated until intensity of the bands are quantified using software. For eg. The length of the synthesized HA polymers remained constant in the agarose gel until the concentrations of HAS1, HAS2 and HAS3 reached 15 μM, 10 μM and 15 μM, respectively. These experiments should repeat.

Author Response

Reviewer #1: Comments:

Wang et al have studied and optimized the expression of isoforms of Hyaluronan synthase (HAS1, HAS2, and HAS3) and also conducted kinetic studies. Finally, authors quantified the expression of three homologous using ELISA. Overall the study is comprehensive, structured and provide a major step in our understanding the hyaluronan biosynthesis which is indispensable and play crucial role in dynamic cellular processes. But shortcomings in the results should be addressed before the accepting it for publication.

Thank you very much for your valuable feedback.

  1. In figure 3 (A-C): Why does author not purify HAS based on their sizes (eg. size exclusion chromatography) after affinity chromatography to remove the nonspecific protein bands seen in the SDS PAGE gels and conducted all the further studies with that fraction of purified protein?

Response: Due to the low expression levels and yields of the recombinant membrane proteins HAS1, HAS2 and HAS3, size exclusion chromatography was less effective for fine purification. As an example shown below in the chromatogram, the peak at 8.21 mL corresponded to aggregated HAS3 protein, while the peak at 11.90 mL represented the detergent solubilized HAS3 protein. Furthermore, since purity of the obtained protein examples was good enough for conducting the protein activity analysis, size exclusion chromatography was chosen avoided.

Size exclusion chromatography profile of the purified HAS3 protein

  1. In figure 3D: The western blot quality is very low for both anti-His and anti-Flag. Authors should provide western blot of higher quality or may use anti-MBL antibody. (Ref: Goentzel et al, 2006)

Response: Thank you very much for your important suggestion. We apologize for the low quality of the western blot for anti-His and anti-Flag. Therefore, we prepared new antibodies and repeated the western blot analysis using anti-His tag and anti-Flag tag antibodies. The new results are shown below.

In our manuscript, Figure 3 was modified as follows:

  1. 3. For kinetic studies, authors should check and report the kinetics of at least one of the isoforms as a control after removing the MBP protein, if solubility does not compromise.

Response: We are truly grateful to you for the kind advice. Initially, we attempted to express HAS1, HAS2 and HAS3 in both insect cells (Sf9) and mammalian cells (HEK293F). However, despite our efforts, the protein expression levels were unexpectedly low, which significantly hindered our ability to obtain the proteins for further study. To address this issue, we introduced a solubility-enhancing tag, MBP, to improve the expression and solubility of HAS in HEK293F cells, which ultimately enabled us to successfully obtain the protein. Additionally, we attempted to remove the MBP tag from the HAS-MBP fusion protein using the TEV protease, but unfortunately, we were not able to obtain the expected HAS proteins. In parallel, we reviewed several relevant studies, which suggest that the presence of the MBP tag does not significantly affect the subsequent activity measurements of the fused protein (Nallamsetty et al., 2006; Pennati et al., 2014; Sarker et al., 2019). As a result, we decided to continue our studies with the recombinant HAS-MBP proteins, opting for further characterization, including conducting kinetic analysis.

Nallamsetty, S.; Waugh, D.S. Solubility-enhancing proteins MBP and NusA play a passive role in the folding of their fusion partners. Protein Expr Purif 2006, 45, 175-182, doi:10.1016/j.pep.2005.06.012.

Pennati, A.; Deng, J.; Galipeau, J. Maltose-binding protein fusion allows for high level bacterial expression and purification of bioactive mammalian cytokine derivatives. PLoS One 2014, 9, e106724, doi:10.1371/journal.pone.0106724.

Sarker, A.; Rathore, A.S.; Gupta, R.D. Evaluation of scFv protein recovery from E. coli by in vitro refolding and mild solubilization process. Microb Cell Fact 2019, 18, 5, doi:10.1186/s12934-019-1053-9.

  1. In figure 4A and 4C: the quality of the gels are very poor and the results deduced from the gels are not validated until intensity of the bands are quantified using software. For eg. The length of the synthesized HA polymers remained constant in the agarose gel until the concentrations of HAS1, HAS2 and HAS3 reached 15 μM, 10 μM and 15 μM, respectively. These experiments should repeat.

Response: Thank you very much for your important suggestion. We performed quantitative analysis of the gel images in Figure 4A-4C using Image J software, and generated the plots with GraphPad software. The experiment, which shows that the length of synthesized HA polymers increases with the concentration of HAS, was repeated three times, with only one of the gel results included in the manuscript.

In our manuscript, Figure 4 was modified as follows:

Figure 4. HA biosynthesis in vitro monitored by agarose gel electrophoresis and Stains-All staining. (A-C) AUtilizing different concentrations (0, 1, 3, 5, 10, 15, 20, 30 μM) of HAS1 (A), HAS2 (B), and HAS3 (C) regulated HA synthesis. H*: HA synthesized from 20 μM HAS digested by hyaluronidase. (D-F) Quantitative analysis of the gel images, corresponding to Figure 4A-4C, respectively, was performed using Image J software and the plots were generated with GraphPad software. Error bars represent SD. (G) Size distribution of HA synthesized by mammalian HAS proteins. M, the molecular weight marker represents HA size standards. (H) Monitoring the time course of HA synthesis by HAS2 in vitro using agarose gel electrophoresis. (I) Catalytic activity depends on the concentration of the substrates UDP-GlcA and UDP-GlcNAc.

Reviewer 2 Report

Comments and Suggestions for Authors

The authors of this study analyze the different biochemical properties of the three human HAS isoforms. The article is well-written and easy to read. The main novelty of this study is the fusion of maltose-binding protein (MBP) to HASs which allows for better purification of human HASs. There are some issues that should be addressed by the authors:

1.       In figure 3D and corresponding text, the authors point out that HAS3 displays higher expression (similarly pointed out at figure 7A), but in the anti-His western blot HAS3 seems to be expressed at the lowest level compared to the other isoforms. The authors should figure out why that is and also comment on why HAS3 is expressed at higher levels.

2.       In figure 4E the size of hyaluronan is gradually increased with time. Can the authors suggest an explanation for this observation?

3.       In the text (lines 283-286) and corresponding figures the authors suggest that HASs display differential affinity for UDP-GlcA. The assay that has been used in this occasion can support such conclusion as it measures free UDP after hyaluronan synthesis has occurred. Thus, this assay does not take into consideration differential reaction or retention times of the substrates by the different HAS isoform. The authors should discuss such aspects.

4.       In figure 7 the authors should further include a WB against GFP-HASs to also support the conclusion related to the first comment above.

5.       In figure 7B the authors should include statistical analyses.

6.       In discussion it is critical to mention that mammalian hyaluronan synthase activity has been studied before (https://www.sciencedirect.com/science/article/pii/S0021925819554540), although in that case mouse HASs have been used. The authors should further discuss any differences or agreements between their study and the one mentioned here.

Author Response

Reviewer #2: Comments:

The authors of this study analyze the different biochemical properties of the three human HAS isoforms. The article is well-written and easy to read. The main novelty of this study is the fusion of maltose-binding protein (MBP) to HASs which allows for better purification of human HASs. There are some issues that should be addressed by the authors:

We are grateful for the comments and valuable suggestions.

  1. In figure 3D and corresponding text, the authors point out that HAS3 displays higher expression (similarly pointed out at figure 7A), but in the anti-His western blot HAS3 seems to be expressed at the lowest level compared to the other isoforms. The authors should figure out why that is and also comment on why HAS3 is expressed at higher levels.

Response: In fact, the expression levels of all the three HAS membrane proteins were extremely low, among which HAS3 behaved relatively highest. To improve the gel/image quality of the anti-His and anti-Flag western blot in our manuscript for better expression analysis, we repeated the western blot assay using the fresh-made anti-His tag and anti-Flag tag antibodies. The results are shown in the Figure 3. Additionally, in the Materials and Methods section, it is mentioned that the HAS1, HAS2 and HAS3 proteins were concentrated to 300 µL, with concentrations of 4.3 mg/mL, 4.0 mg/mL, and 5.7 mg/mL, respectively. Moreover, in Figure 7B, the expression of HAS1-GFP, HAS2-GFP, and HAS3-GFP proteins was further confirmed by Western blot analysis using anti-His and anti-Flag antibodies. These results indicate that, in heterologous expression systems using equal scale of cell culture, the expression level of the recombinant membrane protein HAS3 was generally slightly higher than that of HAS1 and HAS2. This observation is consistent with the distinct expression patterns of the three HAS isoforms, where HAS3 is typically expressed at higher levels in certain physiological and pathological conditions. For example, HAS3 expression is significantly increased in certain cancers, which contributes to the metastatic potential of tumor cells. Thus, the higher expression of HAS3 in these systems may reflect its role in more dynamic cellular processes, such as tumor progression and metastasis (Liu et al., 2001; Bai et al., 2005; Kultti et al., 2014).

Liu, N.; Gao, F.; Han, Z.; Xu, X.; Underhill, C.B.; Zhang, L. Hyaluronan synthase 3 overexpression promotes the growth of TSU prostate cancer cells. Cancer Res 2001, 61, 5207-5214.

Bai, K.J.; Spicer, A.P.; Mascarenhas, M.M.; Yu, L.; Ochoa, C.D.; Garg, H.G.; Quinn, D.A. The role of hyaluronan synthase 3 in ventilator-induced lung injury. Am J Respir Crit Care Med 2005, 172, 92-98, doi:10.1164/rccm.200405-652OC.

Kultti, A.; Zhao, C.; Singha, N.C.; Zimmerman, S.; Osgood, R.J.; Symons, R.; Jiang, P.; Li, X.; Thompson, C.B.; Infante, J.R.; et al. Accumulation of extracellular hyaluronan by hyaluronan synthase 3 promotes tumor growth and modulates the pancreatic cancer microenvironment. Biomed Res Int 2014, 2014, 817613, doi:10.1155/2014/817613.

Figure 3. Expression and purification of the recombinant HAS proteins.

Figure 7. Hyaluronan secretion of HEK293T cells transfected with constructs expressing HAS-GFP fusion proteins.

3.2. Expression and Purification of Recombinant Human HAS1, HAS2 and HAS3

Moreover, the expression of recombinant HAS1-MBP, HAS2-MBP and HAS3-MBP proteins was confirmed by western blot analysis using the Anti-His-tag and Anti-Flag-tag antibodies (Figure 3D), original figures can be found in Supplementary Materials Figure S3. These results indicate that, in heterologous expression systems using equal scale of cell culture, the expression level of the recombinant membrane protein HAS3 is generally slightly higher than that of HAS1 and HAS2. This observation is consistent with the distinct expression patterns of the three HAS isoforms, where HAS3 is typically expressed at higher levels in certain physiological and pathological conditions. For example, HAS3 expression is significantly increased in certain cancers, which contributes to the metastatic potential of tumor cells. Thus, the higher expression of HAS3 in these systems may reflect its role in more dynamic cellular processes, such as tumor progression and metastasis [29-31].

  1. In figure 4E the size of hyaluronan is gradually increased with time. Can the authors suggest an explanation for this observation?

Response: Thank you very much for your crucial comments. Hyaluronan (HA) is a linear glycosaminoglycan composed of (→4)-β-D-GlcA-(1→3)-β-D-GlcNAc(1→) disaccharide repeats. The glycosyltransferase HAS adds glucuronicacid and N-acetylglucosamine into their alternating positions in the chain, using UDP-glucuronic acid (UDP-GlcA) and UDP-N-acetylglucosamine (UDP-GlcNAc) as substrates. HAS transfers the UDP-activated glycosyl unit (the donor sugar) to the nonreducing end of the nascent HA polymer (the acceptor sugar). This reaction generates UDP and elongated HA as reaction products. Therefore, agarose gel electrophoresis revealed that the molecular size of HA generated by HAS2 showed a gradual increase in polymer lengths until 2 hours, after which the polymer length no longer increased, synthesizing HA polymer of a consistent length migrating above a 2 × 106 Da.

n UDP-GlcA + n UDP-GlcNAc → 2n UDP + (-4GlcA-β1,3-GlcNAcβ1-)n

We have followed the reviewer’s suggestion and the corrections have been marked in red in the revised manuscript and showed as below:

3.5. Monitoring the Progression of in vitro HA Biosynthesis Over Time

We further examined whether the molecular size of HA was affected by the incubation times. The cultivation time for synthesizing HA by HAS2 was investigated due to its high production rate of high molecular weight HA. The glycosyltransferase HAS incorporates glucuronic acid and N-acetylglucosamine into alternating positions in the chain, utilizing UDP-glucuronic acid (UDP-GlcA) and UDP-N-acetylglucosamine (UDP-GlcNAc) as substrates. Consequently, as the incubation time increases, the size of HA synthesized by HAS2 gradually increases as well. Agarose gel electrophoresis revealed that the molecular size of HA generated by HAS2 showed a gradual increase in polymer lengths until 2 hours, synthesizing HA polymer of a consistent length migrating above a 2 × 106 Da (Figure 4E), original figures can be found in Supplementary Materials Figure S4. The result demonstrated that the HAS enzymes possessed an intrinsic ability to regulate the size of HA produced.

  1. In the text (lines 283-286) and corresponding figures the authors suggest that HASs display differential affinity for UDP-GlcA. The assay that has been used in this occasion can support such conclusion as it measures free UDP after hyaluronan synthesis has occurred. Thus, this assay does not take into consideration differential reaction or retention times of the substrates by the different HAS isoform. The authors should discuss such aspects.

Response: We are truly grateful to you for the kind advice. We followed the advice and revised the description in manuscript and showed as below:

3.7. Kinetic Analyses of Membrane Proteins HAS1, HAS2 and HAS3

With sufficient reaction time, when the concentration of one substrate (either UDP-GlcA or UDP-GlcNAc) becomes oversaturated, the kinetic analysis of the three membrane proteins is performed with respect to the other substrate. The membrane protein HAS2 exhibited lower Km values for both UDP-GlcA and UDP-GlcNAc than those of HAS1 and HAS3, indicating that HAS2 has higher affinity for UDP-GlcA and UDP-GlcNAc than HAS1 and HAS3. Additionally, all HAS enzymes exhibited lower affinity for UDP-GlcNAc than for UDP-GlcA.

  1. In figure 7 the authors should further include a WB against GFP-HASs to also support the conclusion related to the first comment above.

Response: Thank you very much for your nice comments. We have added Figure 7B to the manuscript as suggested, as shown below:

3.8. HAS1-GFP, HAS2-GFP and HAS3-GFP Overexpressing 293T Cells Secreted Hyaluronan

The human HAS1, HAS2 and HAS3 with C-terminally fused green fluorescence protein (GFP) were transiently expressed in mammalian HEK293T cells. In the cells transfected with equal amounts of plasmids, HAS3-GFP exhibited higher protein expression level compared to HAS1-GFP and HAS2-GFP, as indicated by the overall enhanced fluorescence intensity of the positive cells (Figure 7A). The expression of HAS1-GFP, HAS2-GFP and HAS3-GFP proteins was confirmed by western blot analysis using the Anti-His-tag and Anti-Flag-tag antibodies (Figure 7B). The results showed that the expression level of HAS3-GFP is slightly higher than those of HAS1-GFP and HAS2-GFP, which is consistent with what we observed in the heterogenous expression of HASs-MBP.

Figure 7. Hyaluronan secretion of HEK293T cells transfected with constructs expressing HAS-GFP fusion proteins. (A)The fluorescence intensity of GFP control, HAS1-GFP, HAS2-GFP, and HAS3-GFP fusion proteins transfected into HEK293T cells. (B) The western blot analysis examined the expression of HAS1-GFP, HAS2-GFP, and HAS3-GFP proteins using the Anti-His-tag and Anti-Flag-tag antibodies. (C) Hyaluronan ELISA assay-based quantification of secreted HA. Red: Without substrates addition. Blue: The addition of 2 mM UDP-GlcA and 2 mM UDP-GlcNAc. Error bars represent deviations from the means with n = 3 independent experiments, the asterisks indicate a p-Value < 0.01 (**), or < 0.0001 (****).

  1. In figure 7B the authors should include statistical analyses.

Response: Thanks for the reviewer’s careful comments on our manuscript. We followed the advice and revised Figure 7C in manuscript and showed as below:

Figure 7. Hyaluronan secretion of HEK293T cells transfected with constructs expressing HAS-GFP fusion proteins. (A)The fluorescence intensity of GFP control, HAS1-GFP, HAS2-GFP, and HAS3-GFP fusion proteins transfected into HEK293T cells. (B) The western blot analysis examined the expression of HAS1-GFP, HAS2-GFP, and HAS3-GFP proteins using the Anti-His-tag and Anti-Flag-tag antibodies. (C) Hyaluronan ELISA assay-based quantification of secreted HA. Red: Without substrates addition. Blue: The addition of 2 mM UDP-GlcA and 2 mM UDP-GlcNAc. Error bars represent deviations from the means with n = 3 independent experiments, the asterisks indicate a p-Value < 0.01 (**), or < 0.0001 (****).

  1. In discussion it is critical to mention that mammalian hyaluronan synthase activity has been studied before (https://www.sciencedirect.com/science/article/pii/S0021925819554540), although in that case mouse HASs have been used. The authors should further discuss any differences or agreements between their study and the one mentioned here.

Response: Thank you very much for your kind suggestion. We have followed the reviewer’s suggestion and added a discussion on this aspect in the revised manuscript, which is marked in red as shown below:

  1. Discussion

Naoki and colleagues showed that the three isoforms of hyaluronan synthase in mice exhibit distinct enzymatic activities [40]. In their study, the COS-1 cells were used for low-level expression of HAS proteins, and the enzyme stability, HA extension rate, and kinetics of the two substrates UDP-GlcNAc and UDP-GlcUA were analyzed. Similarly, in our study, we performed enzyme analysis on the three human hyaluronan synthase isoforms, evaluating enzyme stability, elongation rate of HA, chain length, and substrate-dependent kinetics. The key difference in our research was that we investigated the large-scale expression of three recombinant human hyaluronan synthases in HEK293F cells, followed by the in vitro purification of the membrane proteins. Additionally, we quantified hyaluronan levels in the culture medium using a specific hyaluronan ELISA.

Reviewer 3 Report

Comments and Suggestions for Authors

The authors evaluated the differential regulation of hyaluronan synthesis by three isoforms of hyaluronan synthases in mammalian cells. I find the article interesting. However, I have the following comments/ questions:

  1. What is the role of HAS2-AS in HA synthesis?
  2. Authors are suggested to check HAS expression in native gels.
  3. What is the expression level of CD44 on the HEK293 cells transfected with HAS isoforms, and how much HA is attached to CD44?
  4. Authors are suggested to check the HA size in the cell culture medium after transfecting HEK293 cells with HAS isoforms.

Author Response

Reviewer #3:

The authors evaluated the differential regulation of hyaluronan synthesis by three isoforms of hyaluronan synthases in mammalian cells. I find the article interesting. However, I have the following comments/ questions:

  1. What is the role of HAS2-AS in HA synthesis?

Response: Natural antisense mRNA of HAS2 (HAS2-AS) is considered an antisense RNA that is transcribed from the opposite strand of the HAS2 gene, which encodes one of the key enzymes (HAS2) responsible for the synthesis of hyaluronan (Nishida. et al., 1999). HAS2-AS is able to regulate HAS2 mRNA levels and hyaluronan biosynthesis and have an important regulatory role in the control of HAS2, HA biosynthesis, and HA-dependent cell functions in vivo (Chao. et al., 2005). HAS2-AS has a key role in TGFβ- and HAS2-induced breast cancer EMT, migration and acquisition of stemness (Kolliopoulos. et al., 2019). Daryn R. and colleagues show that transcriptional induction of HAS2-AS1 and HAS2 occurs simultaneously in PTCs and suggest that transcription of the antisense RNA stabilizes or augments HAS2 mRNA expression in these cells via RNA/mRNA heteroduplex formation (Michael. et al., 2011). HAS2-AS is a regulatory RNA that modulates the expression of the HAS2 gene, thus playing a crucial role in the synthesis of hyaluronan. By controlling HA levels, HAS2-AS influences a variety of biological functions and can impact disease states, highlighting its importance in cellular regulation and potential therapeutic targets.

Nishida, Y.; Knudson, C.B.; Nietfeld, J.J.; Margulis, A.; Knudson, W. Antisense inhibition of hyaluronan synthase-2 in human articular chondrocytes inhibits proteoglycan retention and matrix assembly. J Biol Chem 1999, 274, 21893-21899, doi:10.1074/jbc.274.31.21893.

Chao, H.; Spicer, A.P. Natural antisense mRNAs to hyaluronan synthase 2 inhibit hyaluronan biosynthesis and cell proliferation. J Biol Chem 2005, 280, 27513-27522, doi:10.1074/jbc.M411544200.

Michael, D.R.; Phillips, A.O.; Krupa, A.; Martin, J.; Redman, J.E.; Altaher, A.; Neville, R.D.; Webber, J.; Kim, M.Y.; Bowen, T. The human hyaluronan synthase 2 (HAS2) gene and its natural antisense RNA exhibit coordinated expression in the renal proximal tubular epithelial cell. J Biol Chem 2011, 286, 19523-19532, doi:10.1074/jbc.M111.233916.

Kolliopoulos, C.; Lin, C.Y.; Heldin, C.H.; Moustakas, A.; Heldin, P. Has2 natural antisense RNA and Hmga2 promote Has2 expression during TGFβ-induced EMT in breast cancer. Matrix Biol 2019, 80, 29-45, doi:10.1016/j.matbio.2018.09.002.

  1. Authors are suggested to check HAS expression in native gels.

Response: Thank you very much for your comments. We followed the suggestion to check the HAS expression in the natural gel, but the result seemed to be less than ideal as shown below. The protein samples tended to aggregate in the loading wells, which seems a common issue for many membrane proteins.

  1. What is the expression level of CD44 on the HEK293 cells transfected with HAS isoforms, and how much HA is attached to CD44?

Response: Thank you very much for your nice comments. CD44 is the main cellular receptor for HA. The binding of HA to CD44 induces conformational changes to the receptor as well as post-translational modifications regulating downstream signaling pathways and, as a result, numerous pathobiological processes including inflammation, wound healing, tumor growth, and metastasis (Karousou. et al., 2017; Yang. et al., 2020). In this manuscript, we primarily investigate the in vitro expression and purification of the three isoforms of human hyaluronan synthases, HAS1, HAS2, and HAS3, and further characterize and compare their enzymatic properties. We acknowledge that understanding the interactions between HAS isoforms, CD44, and HA is crucial for a more comprehensive analysis of their roles in cellular processes. Looking ahead, we plan to conduct further experiments to explore this area in greater detail.

Karousou, E.; Misra, S.; Ghatak, S.; Dobra, K.; Götte, M.; Vigetti, D.; Passi, A.; Karamanos, N.K.; Skandalis, S.S. Roles and targeting of the HAS/hyaluronan/CD44 molecular system in cancer. Matrix Biol 2017, 59, 3-22, doi:10.1016/j.matbio.2016.10.001.

Yang, C.; Sheng, Y.; Shi, X.; Liu, Y.; He, Y.; Du, Y.; Zhang, G.; Gao, F. CD44/HA signaling mediates acquired resistance to a PI3Kα inhibitor. Cell Death Dis 2020, 11, 831, doi:10.1038/s41419-020-03037-0.

  1. Authors are suggested to check the HA size in the cell culture medium after transfecting HEK293 cells with HAS isoforms.

Response: Thanks for your suggestion. We transfected 293T cells with HAS1, HAS2, and HAS3, respectively, and we also used agarose gel electrophoresis to analyze the size of HA secreted in the cell culture medium. Unfortunately, the gel results did not show any bands, possibly due to the low protein expression levels of the HAS isoforms. Therefore, we opted for in vitro catalytic synthesis of HA to determine the size of HA.

Reviewer 4 Report

Comments and Suggestions for Authors

This is an interesting and well-done paper on hyaluronan synthases. The authors were successful in expressing the three mammalian hyaluronan synthases in cells HEK293-F, using an N-terminal Flag tag, 8 × His tag, maltose-binding protein, and a tobacco etch virus (TEV) cleavage site. HEK293-F is a highly transfectable subline derived from the human embryonic kidney 293. The 'F' designation indicates that these cells have been adapted for growth in suspension cultures, and are useful for protein production. So, it seems that it is a good system to express soluble enzymes.

The authors showed the expression of the purified recombinant enzymes and then their activity data. They also presented kinetic data and the sizes of the hyaluronan molecules produced by each isoenzyme.

However, the question that has not been addressed is: do these data reflect the activities of native HAS1, HAS2, and HAS3? In the present paper, the solubility of the enzymes was increased by the Flag-tag, and they were not inserted into a membrane. However, the native enzymes are all integral membrane proteins. One side of the enzymes faces the cytosolic environment, while the other side faces the extracellular environment (as shown in Figure 1). Possibly, the different environments can also affect (or signal) the activities of these enzymes. Would the kinetic data, as well as the size of the synthesized hyaluronan molecules, be the same under native conditions? This point is very relevant and should be addressed (or at least discussed).

Author Response

This is an interesting and well-done paper on hyaluronan synthases. The authors were successful in expressing the three mammalian hyaluronan synthases in cells HEK293-F, using an N-terminal Flag tag, 8 × His tag, maltose-binding protein, and a tobacco etch virus (TEV) cleavage site. HEK293-F is a highly transfectable subline derived from the human embryonic kidney 293. The 'F' designation indicates that these cells have been adapted for growth in suspension cultures, and are useful for protein production. So, it seems that it is a good system to express soluble enzymes.

The authors showed the expression of the purified recombinant enzymes and then their activity data. They also presented kinetic data and the sizes of the hyaluronan molecules produced by each isoenzyme.

We appreciate the nice summary.

However, the question that has not been addressed is: do these data reflect the activities of native HAS1, HAS2, and HAS3? In the present paper, the solubility of the enzymes was increased by the Flag-tag, and they were not inserted into a membrane. However, the native enzymes are all integral membrane proteins. One side of the enzymes faces the cytosolic environment, while the other side faces the extracellular environment (as shown in Figure 1). Possibly, the different environments can also affect (or signal) the activities of these enzymes. Would the kinetic data, as well as the size of the synthesized hyaluronan molecules, be the same under native conditions? This point is very relevant and should be addressed (or at least discussed).

Response: Thank you very much for your important suggestion. The expressed and purified HAS1, HAS2 and HAS3 proteins remained membrane proteins, which were stabilized within the micelles formed by the supplemented detergents or in the lipid nanodiscs (the nanodisc datas were not presented, because the nanodisc-reconstituted proteins were not used in the experiments described in this manuscript for the low yield). These three membrane proteins have been shown capable of maintaining their functional activity while in micelles formed by detergents (Chen. et al., 2022; Maloney et al., 2022).

Chen, W.; Cao, P.; Liu, Y.; Yu, A.; Wang, D.; Chen, L.; Sundarraj, R.; Yuchi, Z.; Gong, Y.; Merzendorfer, H.; et al. Structural basis for directional chitin biosynthesis. Nature 2022, 610, 402-408, doi:10.1038/s41586-022-05244-5.

Maloney, F.P.; Kuklewicz, J.; Corey, R.A.; Bi, Y.; Ho, R.; Mateusiak, L.; Pardon, E.; Steyaert, J.; Stansfeld, P.J.; Zimmer, J. Structure, substrate recognition and initiation of hyaluronan synthase. Nature 2022, 604, 195-201, doi:10.1038/s41586-022-04534-2.

We followed the advice and added the description to the discussion section of the manuscript as follows:

  1. Discussion

Furthermore, while the in vitro conditions enabled us to assess the activity of the membrane proteins, they may not have fully replicated their behavior in native membrane environments. The kinetic data obtained from in vitro experiments, as well as the size of the synthesized HA, may differ from what occurs under in vivo physiological conditions. Thus, in the future, further investigation of HASs under physiological conditions may be useful.

Round 2

Reviewer 1 Report

Comments and Suggestions for Authors

Dear Authors,

Thank you for your thorough revisions and detailed responses to the comments.  I have no further comments or suggestions, and I am confident that this version is suitable for publication.

Congratulations on your excellent work, and I wish you success with your research.

Best regards

Reviewer 4 Report

Comments and Suggestions for Authors

Our main concern about this manuscript was addressed and discussed.